# 'Respond'—A novel approach to healthcare delivery for people seeking asylum

Paola Cinardo[1,2]*, Olivia Farrant[3], Philippa Harris[1,2], Kimberlee Gunn[1], Aileen Ni Chaoilte[1], Humayra Chowdhury[1], Allison Ward[1,4,5], Sarah Eisen[1,2,6], Nicky Longley[1,2]

1 Infection Division, Hospital for Tropical Diseases at University College London NHS Foundation Trust, London, United Kingdom, 2 Department of Clinical Research, London School of Hygiene and Tropical Medicine, London, United Kingdom, 3 Imperial College London NHS Foundation Trust, London, United Kingdom, 4 Royal Free London NHS Foundation Trust, London, United Kingdom, 5 Central and North West London NHS Foundation Trust, London, United Kingdom, 6 Children and Young People's Division, University College London NHS Foundation Trust, London, United Kingdom

's These authors contributed equally to this work.

* Paola.Cinardo@lshtm.ac.uk

## Abstract

People seeking asylum (PSA) often experience complex health needs and barriers to healthcare access, yet no "gold-standard" framework for healthcare delivery exists. From July 2021 to March 2023, the 'Respond' service provided community-based holistic health assessments for PSA in temporary accommodation in North-Central London. This paper aims to describe the experience of the Respond pilot by analysing routinely collected retrospective clinical data and semi-structured interviews with service-users and key stakeholders. 86.2% of those eligible (1497/1736) attended the appointment. The majority of service-users were adults travelling alone (75.1%; 1125/1497) and male (75.9%; 1136/1497), with median age 28 years (IQR 23–36). Thirteen percent were children within 116 family units. Most common countries of origin were Iran (24%, 344/1497), Iraq (11.7%, 168/1497), and Afghanistan (9.5%, 136/1497). At least one health need was identified in 83.2% (1246/1497), of which 19.7% (201/1020) were acute health concerns. Half of all adults (52.6%, 634/1206) and 24.0% of children (29/121) had at least one asymptomatic infection. Mental health concerns were reported by 55.9% (669/1197) of adults. Developmental, behavioural or emotional concerns were raised by parents for 17.2% (26/151) of children. Safety concerns were reported by 14.6% (17/116) of families and 7.9% (94/1184) of adults. Service-users and stakeholders reported a positive experience of the holistic approach. Safety and rapport with staff were identified as key to disclosure of sensitive topics. Challenges were highlighted in provision of care for this population and the importance of cross-sectoral collaboration. We demonstrate high rates of engagement and acceptability of a bespoke, holistic healthcare service for PSA. We identified significant physical and mental health needs, and frequent asymptomatic infection in our population. Proactive assessment, by appropriately

**Data availability statement:** Due to the sensitive nature of the data and the specific vulnerability of the asylum-seeking population, public sharing of the dataset is restricted to avoid potential participant identification. UCLH is the data controller for the dataset. Requests for access to anonymised quantitative data from qualified researchers will be considered on a case-by-case basis. Such requests will be subject to review and approval in accordance with UCLH data governance policies. Researchers seeking access should submit a proposal outlining the intended use of the data, which will be reviewed by the UCLH Infection Division Information Governance representatives acting as the data access committee. The UCLH Infection Division Information Governance can be contacted at this email address uclh.infectiongovernance@nhs.net. Data will be shared only where appropriate approvals are in place and where this is consistent with participant consent, ethical approvals, and applicable legal and regulatory requirements.

**Funding:** The author(s) received no specific funding for this work.

**Competing interests:** The authors have declared that no competing interests exist.

trained staff within dedicated, funded services is vital to address health needs and inequalities for this vulnerable population.

## Introduction

Population mobility is at an all-time high [1], due to globalisation, conflict and climate change, among other complex factors. Modern migration impacts both individual and population health [2]. The World Health Organization (WHO) has recently called for a rights-based approach to health that focuses on the individual migrant and their vulnerability to particular health concerns [3].

In the United Kingdom (UK), the number of people seeking asylum (PSA) has risen dramatically since the COVID-19 pandemic [4]. Provision of equitable health services for PSA remains challenging, with strong evidence for multiple and complex barriers in accessing healthcare [5] and significant challenges for healthcare professionals providing it [6].

Need for changes to healthcare delivery for PSA in the UK is well recognised [7], but, despite pockets of excellent practice around the country [8,9], no consistent approach exists. Accurate population-level data regarding the needs of PSA remains lacking, impeding the design, implementation and delivery of evidence-based, cost effective and appropriate services [10–12]. There is no accepted "gold-standard" framework for healthcare delivery to PSA and only limited national guidance [13–18]. Recent NICE guidance for the care of people experiencing homelessness highlights recommendations of relevance to all inclusion health groups [16], including comprehensive and holistic assessments, longer appointment times, and community-based, trauma-informed environments. Cross-sectoral collaborations integrating primary and secondary care, across health and social care, and with the third sector are recommended.

During the COVID-19 pandemic, an unprecedented number of PSA were placed in "contingency accommodation" across North Central London (NCL) [19]. In July 2021, the Respond service pilot was initiated, providing healthcare to PSA in five primary care practices and six contingency hotels across the London boroughs of Barnet, Camden, Islington, and Haringey. The aim of this paper is to describe the experience of the Respond pilot between 7th September 2021 and 6th March 2023. The objectives of this study were to describe the population accessing the Respond pilot service and its health needs by using retrospective analysis of routinely collected clinical data, and to explore acceptability and feasibility of Respond among service users and providers by conducting semi-structured interviews.

Person-centred terminology is used throughout, with formally adopted definitions cited in Table 1.

### The Respond pilot service

The Respond pilot service was designed in consultation with key stakeholders from the National Healthcare Service (NHS), social care, community and third sector and

**Table 1. Definitions.**

| Definitions | |
|---|---|
| Asylum-seeker (here, Person Seeking Asylum) | An individual who is seeking international protection, has applied for asylum and is awaiting a decision on their asylum application. |
| Refugee | A person who has been granted asylum under national legislation. |
| Unaccompanied Asylum-Seeking Child (UASC) (here, Child/young person seeking asylum – unaccompanied (CYPSAR-U) | A person under 18 who is applying for asylum in the UK in their own right, is separated from both parents, and is not being cared for by a relative or guardian in the UK. |
| Initial accommodation (IA) | Under Section 98 of the Immigration and Asylum Act 1999, if an individual can show they are destitute when they first apply for asylum, they will be provided with 'Initial Accommodation' (IA) while the Home Office assesses their eligibility for longer-term (Section 95) support. IA are typically large full-board hostels with shared bedrooms, living and social areas [20]. |
| Contingency accommodation | Contingency accommodations are used when there are not enough IA [21]. They currently include use of hotels, repurposed Ministry of Defence (MoD) facilities, student and other self-contained accommodation [18]. |
| RHS-13 | The Refugee Health Screener questionnaire is a validated tool for the assessment of mental health in refugees and is included in the Respond health assessment [22,23]. |
| Trauma-informed practice | An approach to health and care interventions which is grounded in the understanding that trauma exposure can impact an individual's neurological, biological, psychological and social development [24]. |

The definition of asylum-seeker, refugee and migrant are from the UNHCR Master Glossary of Terms (https://www.unhcr.org/glossary/). The definition of UASC is UK specific and comes from the gov.uk website.

based on prior experience of developing and delivering an integrated healthcare pathway for Children/Young Persons Seeking Asylum-Unaccompanied (CYPSAR-U) [25].

The pilot offered community-based trauma-informed holistic health assessment and care planning to PSA in contingency accommodation in NCL (Fig 1). This included adults travelling alone and adults and children travelling as a family group. CYPSAR-U were not eligible to be assessed in the Respond pilot as they are offered health assessments on a statutory basis.

All communication (administrative and clinical) was supported by a trained telephone interpreter. A migrant health assessment tool was developed based on existing guidance [12,13] and national priorities [26]. It included demographics, assessments of physical, mental, sexual, dental, and safeguarding needs, relevant migration journey information (S1 Text), alongside screening tests (Table 2). Only those problems appropriate for management in primary care were redirected to the registered General Practitioner (GP) and were prioritised by acuity. Complex cases were supported by a monthly virtual multidisciplinary team meeting (MDT) attended by experts in migrant health, mental health, infectious diseases, paediatrics, safeguarding, social care and school nursing, and those professionals involved in the care of the individual.

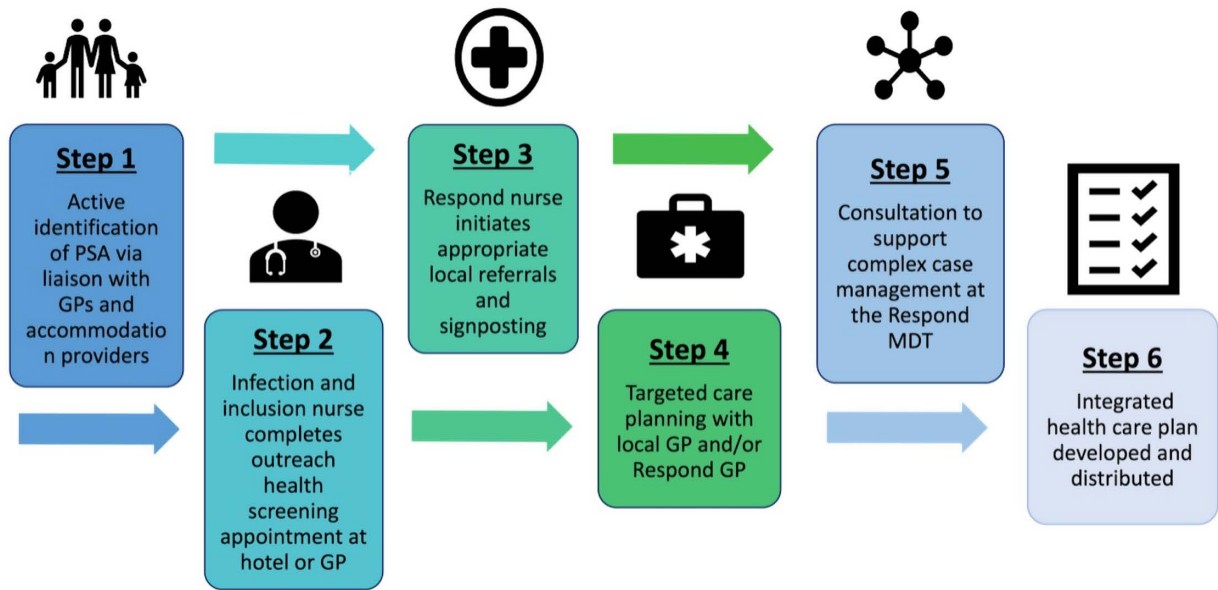

**Fig 1. The Respond pathway.** Step 1: support to register with primary care and appointment offered. Step 2: health needs assessment and screening. Step 3&4: care planning and onward referrals, including signposting, provision of health promotion information, and how to access services. Step 5: advice and guidance MDT to support with complex cases. Step 6: developing of a patient held integrated health care plan, shared with other healthcare professionals, with the aim to ensure continuity of care particularly if PSA are relocated.

## Materials and methods

### Ethics statement

The quantitative data analysis was approved by the Health Research Authority and the Health and Care Research Wales (IRAS 324369, Protocol 158343, REC Reference 23/LO/0858) on 27th August 2024. The interviews were conducted as part of a service evaluation project registered with the University College London Hospital Infection Division Quality Improvement Governance Department. The project adhered to the committee's guidelines, which included assessing the potential impact on participants, ensuring data privacy, and obtaining appropriate consent where required.

### Quantitative data analysis

Clinical data were extracted from University College London Hospital (UCLH) electronic healthcare records (EPIC) into a password protected encrypted.csv file on a secure network. Data were accessed for research purposes on 16th September 2024; they were subsequently cleaned and analysed using descriptive statistics (R version 4.2.1; analysis code available upon request). The authors were involved in the clinical care of the patients and, as such, had access to the medical records of the population; however, data accessed for research purposed were anonymised prior to analysis.

### Clinical data and variables

Extracted data included entries into the assessment tool (S1 Text), Body Mass Index (BMI), blood pressure measurements (mmHg), and results of routine investigations. Any medical issue identified as a result of, or self-reported during, the consultation was recorded under "visit diagnosis", using SNOMED codes in-built in EPIC. Health needs were defined as a) a "visit diagnosis" or b) any abnormal results from the screening assessment listed in Table 2 requiring further action. Where data was missing, results are presented as proportions of available data.

PLOS Global Public
Health

**Table 2. Screening tests conducted during Respond appointment.**

| Screening Assessments | |
|---|---|
| Height, weight and blood pressure* measurement | |
| Blood samples | Full Blood Count & Differential |
| | Liver Profile |
| | Renal* & Bone Profiles |
| | Haemoglobin A1c* |
| | Vitamin D |
| | Quantiferon TB Test (IGRA) |
| | Hepatitis B Full Screen (sAg, sAb, cAb) |
| | Hepatitis C serology |
| | HIV Antibody/ Antigen |
| | Syphilis Antibody Testing |
| | Schistosoma serology |
| | Strongyloides serology |
| *If from Central or South America* | Trypanosoma Cruzi (T. Cruzi) serology |
| Urine | Chlamydia and Gonorrhoea Nucleic Acid Amplification Test (NAAT)* |
| Stool | Stool microscopy for ova, cysts and parasites or Novodiag Stool Parasite Assay** |
| | Entamoeba histolytica, Giardia, cryptosporidium PCR |
| | Helicobacter pylori Antigen[1] |

*In children <18yo, renal profile, HbA1c, and blood pressure were not measured. Chlamydia and Gonorrhoea NAAT was tested in post-pubertal individuals or if deemed clinically appropriate. **Stool microscopy performed until 24.10.22, Novodiag thereafter, due to institutional testing changes.

## Qualitative data

Qualitative data were collected using semi-structured interviews with service users and providers between May-July 2022. The recruitment period started 5th May 2022 and ended 27th July 2022. Inclusion criteria for service users included in the qualitative analysis were: individuals accessing the Respond pilot, > 18 years old, ability to speak English, digitally literate, and score <12 on the Refugee Health Screener-13 (RHS-13) [22]. Service providers were recruited among those providing care for PSA in accommodation served by Respond. Purposive sampling was used for both groups to ensure inclusion criteria were met and to optimise the range of perspectives across key stakeholders (i.e., primary care, secondary care, public health, third sector organisations). The Topic Guide (S2 Text) for semi-structured interviews was developed using the Levesque et al. conceptual framework [27] for assessing healthcare access. Participants were contacted by telephone. Verbal consent was taken after an explanation of the voluntary nature of participation. Interviews lasted 20 minutes, were conducted by three of the authors (OF, PC, KG), and were recorded and transcribed using Microsoft Teams according to institutional processes. Transcriptions were reviewed and anonymised by researchers. To minimise researcher bias, two researchers (OF, KG) independently cleaned and analysed transcripts, with initial familiarisation of the data. Researchers came together fortnightly during analysis to generate codes deductively according to the Levesque et al. framework. Thematic analysis was employed using Braun & Clarke's 2006 six stage model [28]. An inductive approach to data analysis was adopted for service provider interviews, where themes were generated empirically from data with no *a priori* presumptions. For service users, the analytical approach was more deductive, due to the more focussed nature of questions in the topic guide. Codes and themes were recorded and shared using Microsoft Excel.

### Reflexivity

The researchers are white, cis females from high-income countries, with no personal experience of forced migration. We acknowledge that our institutional roles, professional background, experiences, prior assumptions and potential personal investment in outcome may influence the research process; a conscious effort of reflection throughout the project was made to minimise this.

## Results

### The Respond service

A total of 1736 people were identified as eligible to be assessed in the Respond service, and all were offered an appointment. Of those, 86.2% (1497/1736) attended. Of those who attended, 92.2% (1381/1736) attended the first appointment, and 91.6% (1371/1736) consented to investigations (Fig 2). An Integrated Healthcare Plan was created for 97.6% (1461/1497). An interpreter was needed in 73.7% (1103/1497) appointments.

Overall, 67.6% (1012/1736) PSA were assessed in the first 12 months after arrival in the UK, with 40.7% (610/1497) seen in the first 6 months.

### Population characteristics

**Sociodemographic.** Population characteristics are shown in Table 3. Children < 18y constituted 13.2% (197/1497) of our population. Median number of children per family was 1 (range 1–6). Male patients were 75.9% (1136/1497). Median age for adults and children were 30 years (IQR 25–37) and 7 years (IQR 3–11), respectively. Patients were from 74 countries (Fig 3), mainly Iran (344/1497; 24%), Iraq (168/1497; 11.7%), Afghanistan (136/1497; 9.5%, Eritrea (83/1497; 5.8%) and Syria (71/1497; 4.9%).

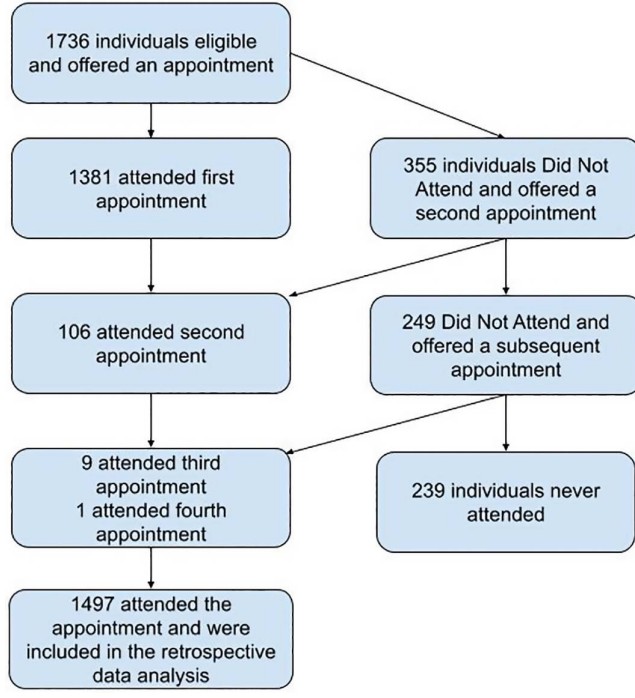

**Fig 2. Flow chart for inclusion in the retrospective analysis.**

**Table 3. Characteristics of the Respond population.**

| | Total (n = 1497, 100%) | Adults (n = 1300, 86.8%) | Children (n = 197, 13.2%) | |
|---|---|---|---|---|
| *Sociodemographic characteristics* | | | | |
| Age y; median (IQR) | 28 (23 –36) | 30 (25 –37) | 7 (3 –11) | |
| Male n (%;, %, 95% CI) | 1136(75.9; 73.7-78) | 1039(79.9; 77.7-82) | 97(49.2; 42.3-56.2) | |
| Country of origin (10 most represented) n (%; 95% CI) | | | | |
| Iran | 344(24.0; 21.8-26.2) | 324(25.6; 23.3-28.1) | Iraq | 23(13.6; 9.2-19.6) |
| Iraq | 168(11.7; 10.1-13.5) | 145(11.5; 9.8-13.3) | UK | 22(13.0; 8.8-18.9) |
| Afghanistan | 136(9.5; 8.1-11.1) | 127(10.0; 8.5-11.8) | Iran | 20(11.8; 7.8-17.6) |
| Eritrea | 83(5.8; 4.7-7.1) | 83(6.6; 5.3-8.1) | Kuwait | 14(8.3; 5-13.4) |
| Syria | 71(4.9; 3.9-6.2) | 68(5.4; 4.3-6.8) | El Salvador | 11(6.5; 3.7-11.3) |
| Sudan | 70(4.9; 3.9-6.1) | 68(5.4; 4.3-6.8) | Afghanistan | 9(5.3; 2.8-9.8) |
| Kuwait | 62(4.3; 3.4-5.5) | 48(3.8; 2.9-5) | Honduras | 8(4.7; 2.4-9.1) |
| El Salvador | 47(3.3; 2.5-4.3) | 36(2.8; 2.1-3.9) | Albania | 6(3.6; 1.6-7.5) |
| Albania | 37(2.6; 1.9-3.5) | 31(2.4; 1.7-3.5) | Bangladesh | 5(3.0; 1.3-6.7) |
| Honduras | 30(2.1; 1.5-3) | 22(1.7; 1.2-2.6) | Pakistan | 5(3.0; 1.3-6.7) |
| Other | 387(27.0; 24.7-29.3) | 314(24.8; 22.5-27.3) | Other | 46(27.2; 21.1-34.4) |
| Available data | 1435(95.8) | 1266(97.4) | 169(85.8) | |
| Language (5 most spoken languages) n (%; 95% CI) | | | | |
| Arabic | 272 (24.1; 21.7-26.7) | | | |
| Kurdish | 237 (21.0; 18.8-23.5) | | | |
| Persian | 175 (15.5; 13.5-17.8) | | | |
| Spanish | 72 (6.4; 5.1-8) | | | |
| Tigrinya | 55 (4.9; 3.8-6.3) | | | |
| Other | 316 (28.0; 25.5-30.7) | | | |
| Available data | 1127 (75.3) | | | |
| Adults travelling alone n (%; 95% CI) | 1125 (75.1; 73.0-77.1) | | | |
| Family unit (n) | 116 | | | |
| Median number of persons per family (range) | 3 (2 –8) | | | |
| Median number of children per family(range) | 1 (1 –6) | | | |
| Family links in the UK prior to arrival n (%, 95% CI) | 265 (20.1; 18-22.4) | | | |
| Available data | 1317(88.0) | | | |
| Travel duration in months n (%; 95% CI) | | | | |
| 0–3 months | 587 (53.3; 50.3-56.2) | | | |
| 3–6 months | 77(7.0; 5.6-8.6) | | | |
| 6–12 months | 73(6.6; 5.3-8.2) | | | |
| +12 months | 365(33.1; 30.4-36) | | | |
| Available data | 1102(73.6) | | | |
| *Behavioural characteristics* | | | | |
| Current smoker n (%; 95% CI) | | 438 (33.7; 33.4-38.8) | | |
| Alcohol use n (%; 95% CI) | | 288 (23.7; 21.4-26.1) | | |
| Available data | | 1217(100) | | |

*(Continued)*

| | Total (n = 1497, 100%) | Adults (n = 1300, 86.8%) | Children (n = 197, 13.2%) |
|---|---|---|---|
| *Anthropometric characteristics* | | | |
| BMI (n = available data) | | n = 874(67.2) | n = 109 (69)£ |
| Obese (BMI > 30) n (%; 95% CI) | | 141 (16.1; 13.8-18.7) | 15 (13.8; 8.5-21.5) |
| Overweight (BMI 25-29.9) n (%; 95% CI) | | 280 (32.0; 29-35.2) | 14 (12.8; 7.8-20.4) |
| Healthy (BMI 18.5 – 24.9) n (%; 95% CI) | | 429 (49.1; 45.8-52.4) | 78 (71.6; 62.5-79.2) |
| Underweight (BMI < 18.5) n (%; 95% CI) | | 24 (2.7; 1.9-4.1) | 2 (1.8; 0.5-6.4) |

IQR: Interquartile range; BMI: Body Mass Index. £ BMI in children was calculated according to the Royal College of Paediatrics and Child Health chart for children >2 years old [29].

Adults travelling alone accounted for 75.1% (1125/1497) of the total population. A third (33.1%; 365/1102) had travelled for more than a year before arrival in the UK. All children were within 116 family groups, half (53.4%; 62/116) with only one parent (Table 3).

**Anthropometric and behavioural characteristics.** Among adults, 48.2% (421/874) were overweight (BMI > 25), with 141/874 (16.1%) classified as obese (BMI > 30). Of the children, 26.6% (29/109) were overweight (BMI > 91st centile) and 13.8% (15/109) were obese. Few (2.7%, 24/874 of adults and 1.8%, 2/109 of children) were underweight (BMI < 18.5 for adults, 2nd centile for children).

Adults who reported being a current smoker were 33.7% (438/1300), and alcohol use was reported by 288/1216 (23.7%) asked.

## Health needs

Health needs are described in Table 4. As shown, 83.2% (1246/1497) of the population had at least one health need identified by the assessment (S1 Text), with 57.8% (865/1497) reporting multiple health needs (i.e., more than one health need). Of those with multiple needs, median of health needs per person was 3 (range 2–10). Regarding children specifically, 31.0% (61/197) had a single health need and 22.8% (45/197) multiple needs.

**Physical health.** An immediate health concern was disclosed by 19.7% (201/1497) individuals. This included rash (13), chest pain (9), cough for more than 2 weeks (6), fever (4), diarrhoea (3), night sweats (2), vomiting (1) and "other" (131).

Previously known medical problems were reported by 16.9% (235/1497) individuals. These included diabetes (17/1497; 1.1%), hypertension (18/1497; 1.2%), asthma (17/1497; 1.1%), and gastrointestinal (40/1497; 2.7%), kidney/urinary (17/1497; 1.1%), neurological (24/1497; 1.6%), musculoskeletal (82/1497; 5.5%) and dermatological (24/1497; 1.6%) problems. Patients with known malignancy constituted 0.3% (5/1497).

Blood pressure values > 140/90 mmHg were found in 18.3% (172/941) patients, who were subsequently referred to primary care to confirm a diagnosis of hypertension.

Low values of haemoglobin using age and gender specific cut-off were observed in 7.2% (84/1206) of adults and 8.8% (9/121) of children. Low vitamin D (< 50 nmol/L) was identified in 78.3% (999/1275) of the Respond population, with 32.3% (558/1170) of adults and 4.3% (15/105) of children suffering from severe deficiency (<25 nmol/L). An HbA1c >= 48 mmol/mol was found in 3.2% (35/1097) of adults, indicating either poorly controlled or a new diagnosis of diabetes.

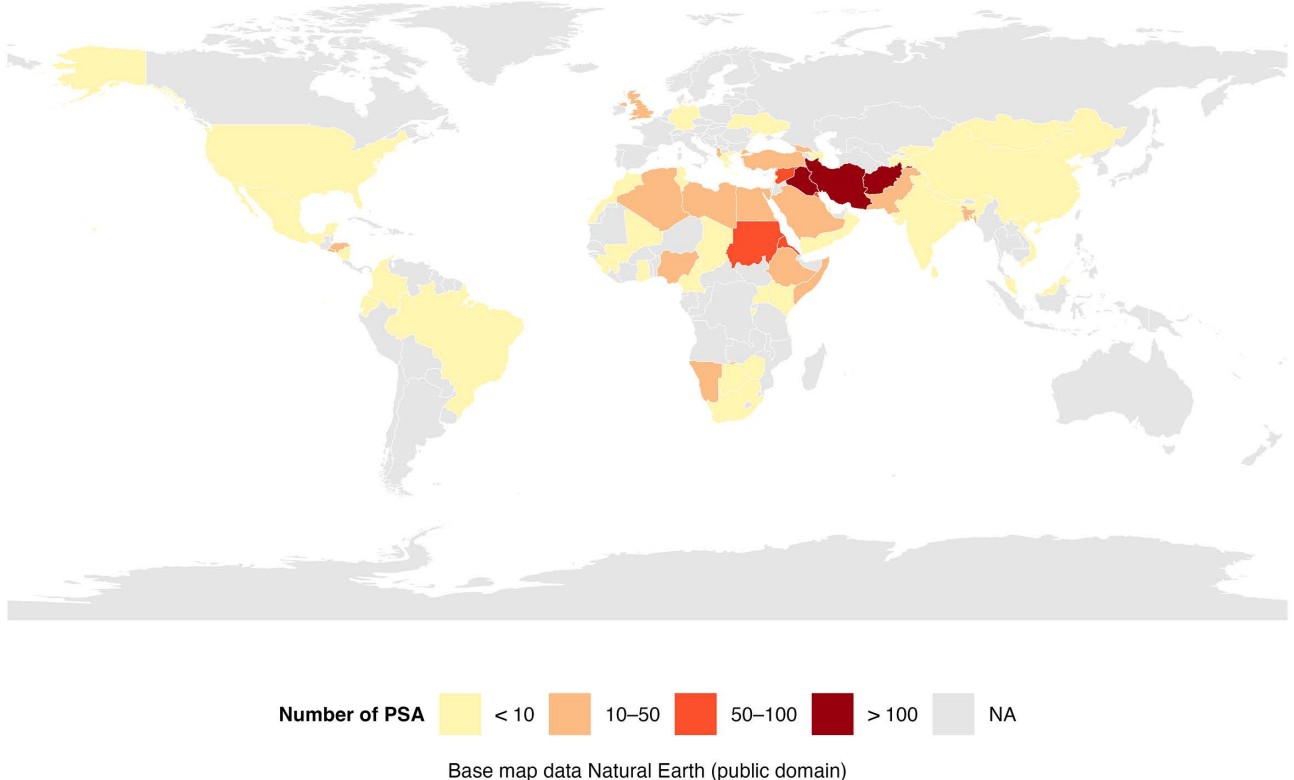

Number of PSA   < 10   10–50   50–100   > 100   NA

Base map data Natural Earth (public domain)

**Fig 3. Map of countries of origin.** The cutoff of 'less than 10 patients' is used to reduce the risk of deductive recognition.

At least one infection was identified in 52.6% (634/1206) of adults and 24% (29/121) of children tested; more than one infection was identified in 16.3% (217/1327) of the total population. All infections identified were referred for treatment (Table 4). Details of infection results will be discussed elsewhere.

**Immunisations.** Previous immunisations were reported by 85% (1098/1292; 95% CI: 82.9-86.8%) individuals; 75.6% (396/524; 95% CI: 71.7-79.1%) of adults and 52.9% (45/85; 95% CI: 42.4-63.2%) of children had no confirmatory documentation. Of the adults, 74.2% (873/1176; 95% CI: 71.7-76.7%) reported that they had received at least one dose of COVID-19 vaccine.

**Women's health needs.** Among women of reproductive age (15-49y, n = 243), 77/243 (31.7%; 95%CI: 26.2-37.8%) had concerns regarding menstruation and 24/243 (11.1%; 95%CI: 7.7-15-7%) were pregnant, with 2 referred into antenatal care by Respond. 80/219 (36.5%; 95%CI: 30.4-43.1%) eligible women (25–64 years old) had not had or did not know if they had had a cervical smear.

**Adult mental health needs.** Of the adults who completed the mental health section of the health assessment (S1 Text), 55.9% (669/1197; 95%CI: 53.1-58.7%) reported a current mental health problem, with thoughts of self-harming reported by 20.1% (224/1117; 95% CI: 17.8-22.5%) of those asked. The Refugee Health Screener 13 was used to assess psychological distress, with a cut off of 11 for individuals that required additional mental health support [30,31]. 55.7% (724/1300; 95% CI: 52.9-58.4%) number of adults had values of RHS 13 above 11, indicating high risk of mental health distress.

**Adult safeguarding needs.** Among the adults, 7.9% (94/1184; 95%CI: 6.5-9.6%) reported feeling unsafe since arrival to the UK. More than a third (36.7%, 95%CI: 33.9-39.5%; 410/1118) disclosed torture, 37.8% (379/1002; 95%CI: 34.9-40.9%)

**Table 4. Health needs of the Respond population.**

| Health needs | Total population (n = 1497; 100%) | Adults (n = 1300; 86.8%) | Children (n = 197; 13.2%) |
|---|---|---|---|
| One health need n (%; 95% CI) | 381 (25.4; 23.3-27.7) | 320 (24.6; 22.4-27) | 61 (31.0; 24.9-37.7) |
| Multiple health needs n (%; 95% CI) | 865(57.8; 55.3-60.3) | 820 (63.1; 60.4-65.7) | 45 (22.8; 17.5-29.2) |
| No needs identified n (%; 95% CI) | 251(16.8; 15-18.7) | 160(12.3; 10.6-14.2) | 91(46.2; 39.4-53.2) |
| Physical health needs | | | |
| Acute health concerns n (%; 95% CI) | 201 (19.7; 17.4-22.3) | 190 (21.2; 18.6-23.9) | 11 (9; 5.1-15.4) |
| Current or known medical conditions n (%; %) | 253 (16.9; 15.1-18.9) | 238 (18.3; 16.3-20-5) | 15 (7.6; 4.7-12.2) |
| High Blood Pressure (BP) n (%; 95% CI) – BP > 140/90 mmHg | | 172 (18.3; 15.9-20.9) | |
| Available data | | 941 (72.4) | |
| Dental pain n (%; 95% CI) | 431 (32.2; 29.8-34.8) | 410 (34.4; 31.8-37.2) | 21 (14.3; 9.5-20.9) |
| Available data | 1338(89.4) | 1191(91.6) | 147(74.6) |
| Vision concerns n (%; 95% CI) | 330 (24.8; 22.5-27.2) | 313 (26.6; 24.1-29.2) | 17 (11.1; 7.1-17.1) |
| Available data | 1331(88.9) | 1178(90.6) | 153(77.7) |
| Hearing concerns n (%; 95% CI) | 79 (6.0; 4.8-7.4) | 77 (6.6; 5.3-8.1) | 2 (1.3; 0.4-4.6) |
| Available data | 1324(88.4) | 1171(90.1) | 153(77.7) |
| Immunisation | | | |
| Reported previous immunisations n (%; 95% CI) | 1098 (85.0; 82.9-86.8) | 951 (83.6; 81.3-85.6) | 147 (95.5; 90.9-97.8) |
| Available data | 1292(86.3) | 1138(87.5) | 154(78.2) |
| Documented previous immunisations n (%, 95% CI) | 168 (27.6; 24.2-31.3) | 128 (24.4; 20.9-28.3) | 40 (47.1; 36.8-57.6) |
| Available data | 609(40.7) | 524(40.3) | 85(43.1) |
| Investigations | Total population (n = 1327)* | Adults (n = 1206)* | Children (n = 121)* |
| Hb below limits (adjusted for age and sex)$ n (%; 95% CI) | 93 (7.3; 6-8.9) | 84 (7.2; 5.8-8.8) | 9 (8.8; 4.7-15.9) |
| Available data | 1274(96.0) | 1172(97.2) | 102 (84.3) |
| Vitamin D n (%, 95% CI) | | | |
| Severe deficiency (vit D < 25 nmol/L) | 393 (30.8; 28.3-33.4) | 378 (32.3; 29.7-35) | 15 (14.3; 8.9-22.2) |
| Insufficiency (vit D 25–50 nmol/L) | 606 (47.5; 44.8-50.3) | 558 (47.7; 44.8-50.6) | 48 (45.7; 36.5-55.2) |
| Available data | 1275(96.1) | 1170(97.0) | 105(86.8) |
| HbA1c n (%; 95% CI) | | | |
| Pre-diabetes (42–47 mmol/mol) | | 48 (4.4; 3.3-5.8) | |
| Diabetes (>= 48 mmol/mol) | | 35 (3.2; 2.3-4.4) | |
| Available data | | 1097(91.0) | |
| Infections | | | |
| *Single infection (n; %)* | 446(33.6) | 424 (35.2) | 22 (18.2) |
| *Multiple infections (n; %)* | 217(16.4) | 210 (17.4) | 7 (5.8) |
| *No infections (n; %)* | 664 (50.0) | 572 (47.4) | 92 (76.0) |
| Positive Interferon gamma release assay (IGRA) n (%; 95% CI) | 153 (12.2; 10.5-14.1) | 152 (13.1; 11.3-15.2) | 1 (1.0; 0.2-5.7) |
| Available data | 1255(94.6) | 1159(96.1) | 96(79.3) |
| HIV-1 antibodies detected n (%; 95% CI) | 5 (0.4; 0.2-0-9) | 5 (0.4; 0.2-1) | 0 (0) |
| Available data | 1282(96.6) | 1179(97.8) | 103(85.1) |
| HbsAg living with HBV n (%, 95% CI)° | 25 (1.9; 1.3-2.9) | 25 (2.1; 1.4-3.1) | 0(0) |

*(Continued)*

**Table 4.** (Continued)

| Health needs | Total population (n = 1497; 100%) | Adults (n = 1300; 86.8%) | Children (n = 197; 13.2%) |
|---|---|---|---|
| *Available data* | 1279(96.4) | 1177(97.6) | 102(84.3) |
| Antibodies anti-HCV detected *n (%; 95% CI)* | 8 (0.6; 0.3-1.2) | 8 (0.7; 0.3-1.3) | 0(0) |
| *Available data* | 1281(96.5) | 1178(97.7) | 103(85.1) |
| Positive treponemal antibodies (CMIA) *n (%, 95% CI)* | 11 (0.9; 0.5-1.5) | 11 (0.9; 0.5-1.7) | 0(0) |
| *Available data* | 1268(95.5) | 1179(97.8) | 89(73.5) |
| Positive chlamydia trachomatis urinary NAAT *n (%, 95% CI)* | 7 (0.7; 0.4-1.5) | 7 (0.8; 0.4-1.6) | 0 (0) |
| *Available data* | 933(70.3) | 922(76.4) | 11(9.1) |
| Neisseria gonorrhoea urinary NAAT *n (%; 95% CI)* | 0(0) | 0(0) | 0(0) |
| *Available data* | 933(70.3) | 922(76.4) | 11(9.1) |
| Schistosoma serology positive *n (%; 95% CI)* | 110 (8.7; 7.3-10.4) | 107 (9.2; 7.7-11) | 3 (3.0; 1-8.4) |
| *Available data* | 1264(95.2) | 1163(96.4) | 99(81.8) |
| Strongyloides serology positive *n (%; 95% CI)* | 42 (3.3; 2.5-4.5) | 40 (3.4; 2.5-4.6) | 2 (2.0; 0.5-6.9) |
| *Available data* | 1266(95.4) | 1165(96.6) | 99(81.8) |
| Helicobacter pylori Ag positive *n (%; 95% CI)* | 483 (54.2; 51-57.5) | 465 (57.6; 54.2-61) | 18 (21.7; 14.2-31.7) |
| *Available data* | 890(67.1) | 807(66.9) | 83(68.6) |
| Stool parasites identified (total)£ | 70 (7.5; 6.0-9.4) | 64 (7.6; 6.0-9.6) | 6 (6.9; 3.2-14.2) |
| Non-pathogenic parasites (n; %) | 91 (9.8; 8.1-11.9) | 87 (10.3; 8.4-12.6) | 4 (4.6; 1.8-11.2) |
| *Available data* | 929(70.0) | 842 (69.8) | 87(71.9) |

*Total population (top row) n = number of individuals consenting to testing; children were not tested if born in the UK and parents did not have any infections. $ low Haemoglobin was classified as: Hb < 130 g/L for a male adult population and < 120 g/L for a female adult population. For children, the appropriate age/gender-based value was used. ° With "HbSAg living with HBV", we intend individuals who tested positive for HbSAg, with active Hepatitis B infection. £Results of stool parasites are reported overall despite the use of two different testing methods (stool microscopy and Novodiag stool parasite assay) during the study period. Further exploration of this is beyond the scope of this article.

physical or sexual abuse and 39.2% (416/1062; 95%CI: 36.3-42.1) trauma during their journey. Of adult women, 12.5% (23/184; 95%CI: 8.5-18.1%) reported experiencing female genital mutilation (FGM).

   **Paediatric needs.** Developmental, behavioural or emotional concerns were reported by parents of 17.2% (26/151; 95%CI: 12–24%) children. Dental issues were reported by 14.3% (21/147; 95%CI: 9.5-20.9%) of the children seen in Respond, and 11.1% (17/153; 95%CI: 7.1-17.1%) and 1.3% (2/152; 95%CI: 0.4-4.6%) had visual or hearing issues respectively. Most school-age children (89.9%, 95%CI: 82.4-94.4%; 89/99) were attending school. Safeguarding concerns were raised for 14.6% (17/116) families, while 56.0% (65/116) were referred to a family support worker. Among children, 8.2% (13/158; 95%CI: 4.9-13.6%) already had an assigned social worker. FGM was reported in one girl (1/75; 1.3%, 95%CI: 0.2-7.2%).

## Acceptability – service users

Fourteen service users were approached and eleven participated. Two of these were unable to complete the interview due to technical difficulties with Microsoft Teams. Participant demographics are described in Table 5.

   Overall, major themes identified from service users interviews were interpersonal skills of the provider, safety, and holistic care.

**Table 5. Demographics characteristics of service users interviewed.**

| Service user | Gender | Age range | Country of origin |
|---|---|---|---|
| Service user 1 | Female | 26-35 | Iran |
| Service user 2 | Male | 46+ | Afghanistan |
| Service user 3 | Male | 26-35 | Iran |
| Service user 4 | Female | 36-45 | Sudan |
| Service user 5 | Male | 36-45 | Egypt |
| Service user 6 | Male | 18-25 | Pakistan |
| Service user 7 | Male | 46+ | Pakistan |
| Service user 8 | Male | 36-45 | Trinidad and Tobago |
| Service user 9 | Male | 26-35 | Country not reported due to identification risk |

**Interpersonal skills of the provider.** The most prominent theme related to communication and rapport between the provider and service user. The importance of being heard and being listened to was highlighted.

*"it's six months that we are in the UK and that day was the first day […] which we face it with […] staff of NHS which was listening to us. She has many questions which were relevant to our issues. And she was patient […] she was perfect"* – service user 2

**Safety.** The perceived kindness and empathy of the provider contributed to a feeling of safety to disclose and discuss sensitive topics.

*"I couldn't say to anyone because I thought that if I said to anyone it can make problem for the persons around me. But in that interview, I tell all of my things to that person. I feel safe to say it"* – service user 1

Environment shaped the perception of emotional safety; for one service user, a primary care environment was perceived safe, as a formal medically designated location. Others, however, preferred to be seen at their accommodation.

**Holistic care.** A holistic model of assessment was appreciated, including discussion of well-being factors such as sense of purpose, diet, exercise, and social integration.

*"Yeah, that was actually very helpful […] they check everything, my blood, my urine test and everything and they asked me really positive questions on a way to help me improving my health, mentally and physically, both so that was quite good"* – service user 7

## Acceptability - service providers

15 service providers were approached and 11 participated, representing all boroughs covered by the service (Table 6). Most participants worked within the NHS, with representation from community/primary care services, secondary care commissioning, third sector and private accommodation providers.

Feedback from service providers was positive. Key themes were workload, challenges in providing care for this population, and importance of collaboration.

**Workload.** Many service providers discussed the significant workload of providing care to this population, because of complexity and need for longer healthcare appointments. GPs emphasised the impact of the Respond service in reducing need for complex consultations in primary care, allowing focus on appropriate specific issues.

**Table 6. Characteristics of the service providers interviewed.**

| Service provider (SP) | Sector | Job role |
|---|---|---|
| SP- 01 | Third sector | GP |
| SP- 02 | Private contractor | Hotel manager |
| SP-03 | Private contractor | Accommodation provider manager |
| SP-04 | Public sector – health | Social worker |
| SP-05 | Private contractor | Accommodation provider |
| SP-06 | Public sector - health | ICS programme manager |
| SP-07 | Public sector - education | School Nurse |
| SP-08 | Public sector - health | Health visitor |
| SP-09 | Third sector | GP |
| SP-10 | Private contractor | Hotel manager |
| SP-11 | Public sector – health | GP |

GP = General practitioner; ICS = Integrated Care System.

*"It's really helpful. […] It just sets up a good starting platform to then continue ongoing management of those patients."* – Service Provider 1

However, one GP observed difficulty in organising follow-up and advocated for a more streamlined referral from Respond to GP practices.

*"It [Respond] saves GP time if you're doing things that would have presented to the GP. But if you're sort of looking for problems that would have never materialised otherwise and then sending them, then that's […] maybe not entirely necessary […] it takes a long time if you're a GP waiting to go home when you suddenly get a letter like that and you have no appointment to book them in and you've never seen the patient"* – Service Provider 11

**Challenges in providing care.** Challenges in providing care to this population were described by providers across all sectors. Many highlighted barriers such as language, inadequate appointment duration, staffing shortages and lack of a trauma-informed environment.

*"we've known for a long time that there is a real gap […] in provision and knowledge in healthcare professionals on how they care for our client base, asylum seekers and refugees."* – Service Provider 9

Fragmented health and social care systems, lack of information sharing between providers and lack of understanding of population needs were highlighted.

*"I think we've struggled a bit there […] because we don't completely understand their needs and also because there's not a system in place where they can be easily identified"* – Service Providers 6

*"When the patient doesn't have an NHS number, doesn't have a GP, you've no birth history. So we really have to set it all out and we had to join up the key actions really that we needed the school to do in April for the school nurse to actually do their job."* – Service Provider 7

One GP emphasised the importance of addressing social and emotional well-being as a priority for this population.

*"I think […] a lot of these […] young men, you know, they're really just needed something to do. They needed a job […] or some kind of physical activity. They needed to learn English. […] they need to have meaningful activity more than anything. […] I would say it somehow that could be a more of a priority. […] I think it's important not to over medicalise people who are fit. – Service Provider 11*

**Collaboration.** There was universal acknowledgement from service providers on the collaborative nature of the service, and benefits of working across traditional sectoral boundaries to patient care and staff experience.

*"…we did have some really complex cases that school nurses were dealing with and it was really, really helpful to know and piece together what was actually being done for those children." – Service Provider 7*

*"So now I think it has been very, very helpful. With the previous hotel which I worked in, we didn't have the Respond team and now you can definitely feel it makes a good impact, especially on the staff as well. It just felt like you're sort of being supported in a way." – Service Provider 3*

## Discussion

We describe the outcomes and the experience of service user and providers of a holistic outreach pilot service for PSA living in contingency accommodation in North London. We demonstrate high engagement with the service, significant health needs within the population, and acceptability to both service users and providers.

Among PSA invited to attend the Respond service, 86% attended their appointment, and 92.6% provided samples for screening. This represents higher attendance rates among a population often considered as "hard to reach" [32] compared with similar interventions elsewhere [33,34]. This population faces well-recognised barriers to healthcare access [35,36], including organisational and administrative challenges, cultural and language barriers, limited knowledge about existing services, restricted transportation, and low health literacy [5,36–38]. Additionally, both previous research and our work highlight challenges from the perspective of service providers [6], including need for interpreters, extended appointment time, and trained staff [6].

Although some recommendations exist [13,16,17,39,40], there is no standard framework to deliver care for PSA. However, key principles of good care to address these barriers have been described in the literature [35,41]. These include organisational flexibility, allocation of sufficient time and resources, use of interpreting service, promotion of cultural awareness, collaboration with other services, and engagement in targeted outreach activities [35,41]. The high attendance rates observed in the Respond service may be linked to the implementation of these strategies, all of which were prioritised within the service model. Bespoke organisational processes allowed identification of eligible patients in the community, without the need for referral. Appointments were booked via individual telephone contact using interpreters, and reminders were sent 48 hours before the appointment. The high attendance rate at first appointments in our pilot suggests that the resource required to deliver this proactive approach may be offset by reducing missed appointments.

Appointments included the use of interpreters and were of extended duration. Additional time was allocated to comprehensive care planning to ensure streamlined and appropriate onward referrals with targeted actions. Families were seen together, and care provided in close collaboration with social services. Interviews with service providers highlighted the value of multi-agency collaboration in enhancing service delivery. The community-based setting may also have contributed to engagement. Outreach settings are known to improve uptake of healthcare interventions in socially marginalised groups [42] and are recommended for other "inclusion health" populations in the UK [16].

The Respond pilot was developed as a "one-stop shop", offering comprehensive health assessment and screening tests within a single appointment. This holistic approach, centred around the needs of this population, contrasts with those described elsewhere, which often address either acute presentations [43–46], or a single issue (e.g., infections or mental

health) [47,48]. While our proactive and comprehensive service structure may require more resources and increase the identification of previously unreported problems, early detection and intervention could ultimately reduce unscheduled healthcare visits. Migrant populations are known to have higher rates of unscheduled presentations compared to host populations [49,50] and early intervention may thus have beneficial implications for both individuals and health services [51].

Despite obvious benefits of early identification and treatment of health issues, resource limitations posed significant challenges during the Respond pilot. Interviews with service providers highlighted the fragmented and under-resourced nature of downstream services, particularly in primary care and mental health. Within the pilot, this was, to some extent, mitigated by close cross-sectoral collaboration with stakeholders across the multidisciplinary team. This was recognised by providers as one of the strengths of the Respond service. Similar approaches have been recommended by the WHO in their global plan to improve refugee health [52], and previous research emphasises the importance of multi-agency collaboration in delivering care to PSA [53,54]. We have demonstrated elsewhere that the Respond MDT has contributed to delivery of effective and efficient person-centred, trauma informed care and improved cross-sectoral collaboration [55].

Perhaps the most significant evidence gap in service development for PSA is the lack of the voices of the people with lived experience themselves. Few studies have asked service users "what matters to you?" or evaluated acceptability of the services provided [56,57]. Our interviews showed that a holistic approach is not only acceptable but preferred by service users. Previous studies support this finding [57] and suggest that services focussing on a single element of health, such as infection screening, may be seen as stigmatising [58,59]. Feedback from service users emphasised the importance of appropriately trained trauma-informed healthcare professionals who demonstrate compassion and respect. This approach fosters a sense of 'being heard' and safety, facilitating disclosure of health issues, thus reducing barriers to healthcare access [32].

Our cohort were predominantly male adults travelling alone. We assessed 116 family units, including 197 children of primary school age; adolescents are more likely to migrate without family, and are thus looked after through statutory processes and ineligible for the pilot service. The 3 most represented countries of origin were Iran, Iraq and Afghanistan. Our demographic data reflect national statistics for PSA in the UK and European countries [4,60].

Despite relatively young median age of adults, a high proportion had multiple health needs, consistent with previous studies [61,62] and likely resulting from multiple factors before, during and after migration. This underpins the importance of a holistic approach.

We identified significant numbers of adults with undiagnosed, asymptomatic conditions (such as diabetes and hypertension). In addition to reducing risk of long-term individual morbidity and mortality, early identification of these conditions aligns with national approaches to address health inequalities [26]. These risks may be further mitigated by linkage to care, provided by the pilot with direct referral into appropriate services (and primary care) for management of chronic long-term conditions.

We identified asymptomatic infections in half of the patients tested, with one third of those having multiple infections. This is similar to previous studies [43,47,63]. Helicobacter Pylori and latent tuberculosis infections were particularly prevalent in adults. Number of infections in children was low, especially if compared with rates of infections in older adolescent cohorts [25,64]. Risk of acquiring communicable diseases is high in this population due to prevalence in country of origin, lack of access to healthcare, difficult, prolonged journeys, poor living conditions and malnutrition [25,65,66]. Our results are consistent with previous findings [47,67–70]. Given that infections identified were asymptomatic, and of personal and public health significance, particularly TB and hepatitis B, identification and prompt treatment represent an important opportunity to limit transmission risk [71].

Low rates of confirmed immunisation are concerning, given that PSA living in contingency accommodation in the UK experience risk of overcrowding and consequent outbreaks of vaccine-preventable diseases such as diphtheria, measles or varicella [72,73]. Current recommendation is that those without complete documentation require catch up immunisation

[74,75], which risks a considerable burden on primary care. Further, immunisation has been shown to be more acceptable by marginalised populations if delivered in an outreach setting [76,77]. Aligned with the principle of 'Making Every Contact Count' [78], commissioning of services such as the Respond pilot to commence catch-up immunisations could partially address these challenges [76,77].

Prevalence of self-reported mental health needs was high, affecting half of adults. While it is known that PSA have a higher burden of mental health illness compared to the population of host countries [33], reported rates vary significantly [43,61], likely due to differences in the study population and design, and context of host country [33]. Further research is needed to explore the relationship between self-reported symptoms of mental distress by this population and a diagnosis of mental health illness. Although the optimal timing and intervention for addressing mental health needs in this population remain uncertain [79,80], and access to treatment may be delayed [81,82], early recognition of mental and emotional distress is crucial to allow stabilisation interventions to be initiated [79,83].

Reported safety concerns were identified in 7.9% of our adult cohort and 14.6% of families. PSA are recognised to be at high risk of exploitation [84] and proactive identification and support of those at risk is of paramount importance. Our work also identified a particular need among families, who may benefit from additional support and of whom half were referred to a family support worker.

Our work provides important information about children and young people seeking asylum accompanied by their families (CYPSAR-A), who are less easy to identify and less well characterised than their unaccompanied counterparts [85]. A third of children had an unmet health need, most commonly behavioural or emotional, or a need for additional family support, reaching safeguarding threshold in a significant minority. Dental and visual concerns were prevalent. Children were more likely to be overweight than underweight, perhaps reflecting nutritional challenges during migration journeys and in hotel accommodation. This is consistent with other studies, which show significant unmet educational, safeguarding, mental, physical, and nutritional needs [85–87].

## Limitations

We used routinely collected data and, therefore, data were sometimes incomplete. Our work was conducted in a specific setting, in North Central London, with good access to tertiary specialty services and many third sector organisations, and findings cannot be generalised to PSA populations elsewhere, within whom demographics may also vary geographically. Despite recognition of need [35,41], there remains no "gold standard" framework against which to evaluate inclusion health services, and thus conclusions remain, to some extent subjective and of limited generalisability [35,41].The qualitative analysis was limited as this work was a service evaluation rather than qualitative research. Additionally, interviews with service users included, for practical reasons, only those who could speak English and who had digital literacy and access. This risks bias towards a subgroup with longer duration in the UK and of educational and economic circumstances which may not be representative of all service users. Most importantly, it was not possible to further include the voices of the population themselves in designing, implementing and analysing this work. Patient engagement work is underway, including detailed exploration of the views of service users on the assessment questionnaire.

## Conclusions and future research

To our knowledge, we present the largest dataset of clinically collected data describing health needs of PSA in the UK to date. Health needs have been previously described in the UK and other high-income countries [5,62,88], but data are often based on specific aspects such as communicable diseases or mental health [47,89] with few studies reporting broader needs holistically [32,90]. We demonstrate diverse, extensive and complex health needs, and benefit of bespoke, holistic and collaborative services to address healthcare barriers for this population [32].

We demonstrate that an outreach health assessment service for PSA is acceptable, feasible and deliverable in this setting. The Respond assessment approach is adaptable and can be delivered in a range of contexts. Further work will focus

on exploration of longer-term impact on individuals and linkage to care and outcomes, and on health systems, including economic analysis. Implementation and evaluation of an upscaled and hub-spoke model, linking outreach services to centres of expertise, will also be explored [91]. Most importantly, formal research and co-creation with service users is planned, to ensure that the voice of this marginalised population remains central to decisions made about their care.

## Supporting information

**S1 Text. Holistic health needs questionnaire.**
(DOCX)

**S2 Text. Service evaluation topic guide.**
(DOCX)

## Acknowledgments

We would like to thank the entire Respond team—particularly the nurses, clinicians, administrators and managers — whose commitment, professionalism and dedication made this service possible.

We are immensely grateful to the individuals and families seeking asylum who participated in this evaluation. Their willingness to share their experiences has been invaluable.

Finally, we acknowledge the contributions of local authority teams and voluntary sector organisations, whose collaboration was extremely valuable in delivering the service.

## Author contributions

**Conceptualization:** Paola Cinardo, Olivia Farrant, Allison Ward, Sarah Eisen, Nicky Longley.

**Data curation:** Humayra Chowdhury.

**Formal analysis:** Paola Cinardo, Kimberlee Gunn.

**Methodology:** Paola Cinardo, Olivia Farrant, Kimberlee Gunn, Sarah Eisen, Nicky Longley.

**Validation:** Philippa Harris.

**Visualization:** Paola Cinardo.

**Writing – original draft:** Paola Cinardo, Olivia Farrant.

**Writing – review & editing:** Paola Cinardo, Philippa Harris, Kimberlee Gunn, Aileen Ni Chaoilte, Allison Ward, Sarah Eisen, Nicky Longley.

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
