## [Decision Letter · Decision Letter 0]

9 Sep 2025

PGPH-D-25-01755

‘Respond’ - a novel model of healthcare delivery for people seeking asylum

Dear Dr. Cinardo,

Thank you for submitting your manuscript to PLOS Global Public Health. After careful consideration, we feel that it has merit but does not fully meet PLOS Global Public Health’s publication criteria as it currently stands. Therefore, we invite you to submit a revised version of the manuscript that addresses the points raised during the review process.

The reviewers have made very useful suggestions on how to improve the manuscript. I invite the authors to consider the reviewers' comments and address them meticulously.

We look forward to receiving your revised manuscript.

Kind regards,

Ferdinand C Mukumbang, PhD

Academic Editor

Journal Requirements:

1.  Please send a completed 'Competing Interests' statement, including any COIs declared by your co-authors. If you have no competing interests to declare, please state "The authors have declared that no competing interests exist". Otherwise please declare all competing interests beginning with the statement "I have read the journal's policy and the authors of this manuscript have the following competing interests:"

2. In the online submission form, you indicated that Due to the sensitive nature of the data and the specific vulnerability of the asylum-seeking population, public sharing of the dataset is restricted to avoid potential participant identification. However, de-identified data supporting the findings of this study are available from the corresponding author upon reasonable request.

3. Uploaded as supplementary information.

3. We do not publish any copyright or trademark symbols that usually accompany proprietary names, eg (R), (C), or TM  (e.g. next to drug or reagent names). Please remove all instances of trademark/copyright symbols throughout the text, including ©, ™ on page 7, 8, 9.

4. Some material included in your submission may be copyrighted. According to PLOS’s copyright policy, authors who use figures or other material (e.g., graphics, clipart, maps) from another author or copyright holder must demonstrate or obtain permission to publish this material under the Creative Commons Attribution 4.0 International (CC BY 4.0) License used by PLOS journals. Please closely review the details of PLOS’s copyright requirements here: PLOS Licenses and Copyright. If you need to request permissions from a copyright holder, you may use PLOS's Copyright Content Permission form.

Potential Copyright Issues:

Figure 2: please (a) provide a direct link to the base layer of the map (i.e., the country or region border shape) and ensure this is also included in the figure legend; and (b) provide a link to the terms of use / license information for the base layer image or shapefile. We cannot publish proprietary or copyrighted maps (e.g. Google Maps, Mapquest) and the terms of use for your map base layer must be compatible with our CC-BY 4.0 license.

5. We have noticed that you have uploaded Supporting Information files, but you have not included a list of legends. Please add a full list of legends for your Supporting Information files after the references list.

Reviewers' comments:

Reviewer's Responses to Questions

**Comments to the Author**

1. Does this manuscript meet PLOS Global Public Health’s publication criteria? Is the manuscript technically sound, and do the data support the conclusions? The manuscript must describe methodologically and ethically rigorous research with conclusions that are appropriately drawn based on the data presented.? Is the manuscript technically sound, and do the data support the conclusions? The manuscript must describe methodologically and ethically rigorous research with conclusions that are appropriately drawn based on the data presented.

Reviewer #1: Yes

Reviewer #2: Yes

2. Has the statistical analysis been performed appropriately and rigorously?

Reviewer #1: Yes

Reviewer #2: Yes

3. Have the authors made all data underlying the findings in their manuscript fully available (please refer to the Data Availability Statement at the start of the manuscript PDF file)?

The PLOS Data policy requires authors to make all data underlying the findings described in their manuscript fully available without restriction, with rare exception. The data should be provided as part of the manuscript or its supporting information, or deposited to a public repository. For example, in addition to summary statistics, the data points behind means, medians and variance measures should be available. If there are restrictions on publicly sharing data—e.g. participant privacy or use of data from a third party—those must be specified.requires authors to make all data underlying the findings described in their manuscript fully available without restriction, with rare exception. The data should be provided as part of the manuscript or its supporting information, or deposited to a public repository. For example, in addition to summary statistics, the data points behind means, medians and variance measures should be available. If there are restrictions on publicly sharing data—e.g. participant privacy or use of data from a third party—those must be specified.

Reviewer #1: No

Reviewer #2: Yes

4. Is the manuscript presented in an intelligible fashion and written in standard English?

Reviewer #1: Yes

Reviewer #2: Yes

Reviewer #1: This manuscript reports on the Respond pilot, a community-based health assessment model for people seeking asylum in North Central London. The topic is timely and relevant and the dataset is substantial. My overall recommendation is major revision.

Suggestions for improvement -

Clarity of study design and objectives (Lines 42–47, 118–125, 146–166)

Please make the objectives clearer. At present, it is not obvious whether the study is primarily descriptive (health needs), an assessment of feasibility, or an evaluation of service impact. Kindly edit the introduction and methods to set out primary and secondary aims more explicitly. It may also help to align the reporting with STROBE (quantitative) and COREQ (qualitative).

Ethical approval timing (Lines 167–188)

There appears to be a mismatch between the timing of ethics approval (August 2024) and the interview period (May–July 2022). Please clarify how interviews were governed ethically, including how consent, confidentiality and safeguarding were ensured. Kindly check if this information is already in the methods and expand if needed.

Sampling and selection bias (Lines 189–200; 300–309)

For the quantitative analysis: Kindly add a simple flow diagram showing the pathway from eligibility to attendance and inclusion. This would make the sample derivation transparent.

For the qualitative analysis: Restricting to English-speaking, digitally literate participants with low RHS-13 scores may bias the sample. Please acknowledge this limitation more clearly in the text.

Measurement issues (Lines 240–279)

Mental health: Please state exactly which tool was used for routine assessment, the cut-offs applied and whether interpreter support was available.

Women’s health: Kindly correct the denominator for cervical screening, since in the UK the eligible age is 25–64 rather than 15–49.

Vitamin D: Please convert results from ng/mL to nmol/L and use standard cut-offs.

HbA1c: Kindly report in mmol/mol rather than “mmol.”

Stool testing: The switch from microscopy to Novodiag (Line 143) introduces heterogeneity. Please highlight this in the limitations.

Statistical analysis (Lines 146–167; 205–239)

The results are entirely descriptive. Kindly add confidence intervals for key proportions.

Linkage to care and outcomes (Lines 244–299)

The discussion suggests improved linkage to care, but no data are shown.

Qualitative analysis transparency (Lines 168–188; 300–336; 340–397)

Kindly expand the description of how coding and theme development were conducted, whether saturation was reached and how disagreements were resolved. Please also consider adding a table that links themes with quotes for clarity.

Minor suggestions for improvement

• The abstract would benefit from being restructured in the standard PLOS format, with clear sections for Background, Methods, Findings and Interpretation.

• Throughout the manuscript there are several instances of “Error! Reference source not found.”

• Terminology is used inconsistently across the paper—different acronyms such as PSA, PSAR, CYPSAR-U and CYPSAR-A appear.

• Please also standardise the units to UK conventions: mmol/mol for HbA1c, nmol/L for vitamin D and mmHg for blood pressure. In the women’s health section, the phrase “living with HBV” should be clarified as “surface antigen positive,” and it would be helpful to note the follow-up process.

Reviewer #2: I read Respond's evaluation very carefully. The article describes the importance of participation in assessing the components involved in healthcare delivery for people seeking asylum.

I draw on what was stated in the Limitations section: "findings cannot be generalized to PSA populations elsewhere, within whom demographics may also vary geographically." I comment that it is inconsistent to call it an evaluation model, and even worse, a care new model, because a model is precisely something to be generalized.

I suggest changing the name "model" to "experience" because, additionally, sharing a good, comprehensive, or holistic experience seems like an invitation to observe and even adapt it to other conditions and realities, which would be the goal of this article.

**Do you want your identity to be public for this peer review?** For information about this choice, including consent withdrawal, please see our Privacy Policy..

Reviewer #1: No

Reviewer #2: No

---

## [Decision Letter · Decision Letter 1]

26 Feb 2026

‘Respond’ - a novel approach of healthcare delivery for people seeking asylum

PGPH-D-25-01755R1

Dear Dr Cinardo,

We are pleased to inform you that your manuscript '‘Respond’ - a novel approach of healthcare delivery for people seeking asylum' has been provisionally accepted for publication in PLOS Global Public Health.

Best regards,

Julia Robinson

Executive Editor

Reviewer Comments (if any, and for reference):

Reviewer's Responses to Questions

**Comments to the Author**

Reviewer #2: All comments have been addressed

Reviewer #3: All comments have been addressed

publication criteria? Is the manuscript technically sound, and do the data support the conclusions? The manuscript must describe methodologically and ethically rigorous research with conclusions that are appropriately drawn based on the data presented.? Is the manuscript technically sound, and do the data support the conclusions? The manuscript must describe methodologically and ethically rigorous research with conclusions that are appropriately drawn based on the data presented.

Reviewer #2: Yes

Reviewer #3: Yes

3. Has the statistical analysis been performed appropriately and rigorously?

Reviewer #2: Yes

Reviewer #3: Yes

4. Have the authors made all data underlying the findings in their manuscript fully available (please refer to the Data Availability Statement at the start of the manuscript PDF file)?

The PLOS Data policy requires authors to make all data underlying the findings described in their manuscript fully available without restriction, with rare exception. The data should be provided as part of the manuscript or its supporting information, or deposited to a public repository. For example, in addition to summary statistics, the data points behind means, medians and variance measures should be available. If there are restrictions on publicly sharing data—e.g. participant privacy or use of data from a third party—those must be specified.requires authors to make all data underlying the findings described in their manuscript fully available without restriction, with rare exception. The data should be provided as part of the manuscript or its supporting information, or deposited to a public repository. For example, in addition to summary statistics, the data points behind means, medians and variance measures should be available. If there are restrictions on publicly sharing data—e.g. participant privacy or use of data from a third party—those must be specified.

Reviewer #2: Yes

Reviewer #3: Yes

5. Is the manuscript presented in an intelligible fashion and written in standard English?

Reviewer #2: Yes

Reviewer #3: Yes

Reviewer #2: I congratulate the authors on the changes made to the article ‘Respond’ - a novel approach of healthcare delivery for people seeking asylum. The change in the title and the other sections of the article have made it more coherent and presented it as a shared experience and an invitation to broaden community participation for asylum seekers.

Reviewer #3: Thank you

**Do you want your identity to be public for this peer review?** For information about this choice, including consent withdrawal, please see our Privacy Policy..

Reviewer #2: No

Reviewer #3: **Yes:** Alejandro Gil SalmerónAlejandro Gil SalmerónAlejandro Gil SalmerónAlejandro Gil Salmerón
